# Enhancing Food Security through Digital Inclusive Finance: Evidence from Agricultural Enterprises in China

**DOI:** 10.3390/ijerph20042956

**Published:** 2023-02-08

**Authors:** Siqi Huang, Nik Hadiyan Nik Azman

**Affiliations:** School of Management, Universiti Sains Malaysia, Minden 11800, Penang, Malaysia

**Keywords:** food security, digital inclusive finance, agricultural operating income, food quality and quantity, food wastage

## Abstract

As a means of enhancing food security, efficient agricultural processing and the maintenance of a smooth supply chain are essential for ensuring food quality and reducing food wastage. Agricultural enterprises play a crucial role in the processing and transportation of food from farms to dinner tables. Operating income growth plays the vital role of ensuring that agricultural enterprises function in a stable manner while also indicating the quantity and quality of market food supply. Therefore, the objective of this study is to explore the impact of digital inclusive finance on food security by analyzing the effect of digital inclusive finance on the operating income of agricultural enterprises in China. By applying pooled OLS analysis to Chinese agricultural enterprises that are listed in the National Equities Exchange and Quotations, this study finds that digital inclusive finance can help improve agricultural operating income. The results reveal that digital inclusive finance can facilitate the promotion of agricultural operating income by increasing the supply of financing, accelerating inventory liquidity, and supporting investment in research and development. In addition, this study concludes that digital inclusive finance is more effective for increasing agricultural operating income as a result of its wider coverage and deeper utilization. Furthermore, the development of traditional finance is still necessary for the digitization of digital inclusive finance to be effective.

## 1. Introduction

Ensuring food security is one of the United Nations 17 Sustainable Development Goals (SDGs) and it is the foundation of socioeconomic development. In addition to production, food sales are essential for maintaining a sufficient food supply. Due to many food products’ characteristics of a short shelf life and high storage-related difficulty, timely and unimpeded sales are vital for reducing food waste and ensuring food security [1,2]. However, global food market stability is still challenged by factors such as climate variability, rising agricultural input costs, geopolitical conflicts, and transportation constraints caused by the COVID-19 pandemic [3]. In the process of delivering agricultural products from farmland to consumers, agricultural enterprises play an intermediary role [4,5]. They collect raw agricultural products from farmers and reproduce them in order to provide multiple types of food to consumers. Considering the impact on food security, this study focuses on the operating income of agricultural enterprises in particular. Operating income is the amount that is gained in revenue following the deduction of operating expenses. Agricultural operating income measures the agricultural profit earned by enterprises solely from an operational perspective. As the foundation of profitability and sustainable operation, operating income determines whether an enterprise will remain in the agricultural supply chain, thereby impacting food security [6]. Furthermore, operating income stems from sales revenue, which is based on selling volume and sales price. The improvement of selling volume indicates that consumers can obtain enough goods from the food market, while the ability to sell products at a better price indicates that such products are of satisfactory quality. These two factors are also key determinants of the operating income of agricultural enterprises that can persistently influence their operations [7,8]. Therefore, the growth of operating income is essential for maintaining the sustainable operation of agricultural enterprises and enhancing food security.

According to the Ministry of Industry and Information Technology (MIIT), there are over 40 million enterprises in China, of which 95% are small and medium-sized enterprises [9]. Agricultural enterprises account for just 5.8% of all enterprises in China, of which the vast majority are small and medium-sized enterprises [10]. This is consistent with the overall small scale of agricultural operations, which was reflected in the third agricultural census. However, although they are relatively small in size, these agricultural enterprises play a significant role in maintaining food security. Therefore, the focus of this study is on the operating income of small and medium-sized enterprises in China. Considering the reliability of financial data, this study investigates all agricultural enterprises that are listed in the National Equities Exchange and Quotations (NEEQs) as they are required to provide audited financial statements. Compared to A share-listed agricultural companies, the relatively small scale of the enterprises listed in the NEEQs is more representative. Moreover, these enterprises have supply chains covering both rural and urban areas in China, which can help improve our understanding of the condition of the entire supply chain.

Inclusive finance, which was initially proposed by the United Nations in 2005, refers to universal access to reliable financial services at a reasonable price. The objective in this regard is to improve the provision of financial services to both individuals and micro-, small, and medium-sized enterprises [11]. Inclusive finance in China has progressed to digital inclusive finance due to rapid progress in big data technology. By applying information technologies to finance, digital inclusive finance in China enables enterprises and individuals to obtain financial services at an affordable price [12] and influences consumption and business modes [13,14]. Previous studies have reported that digital inclusive finance can effectively lower financial transaction costs [15,16,17] while providing more financial products [18,19,20]. In addition to financial support, digital finance inclusive is examined as a means of aiding the promotion of agricultural green total factor productivity [21,22] and e-commerce in rural areas [23]. However, due to unbalanced levels of urban and rural development [24], it remains unclear whether digital inclusive finance supports agricultural businesses [25]. Thus, it is meaningful to investigate the effect of digital inclusive finance on the sales of agricultural products in order to better maintain smooth food supply chains covering both urban and rural areas.

Understanding how inclusive digital finance affects agricultural operational revenue will help China maintain food security. Other developing countries can use such knowledge as a model to better maintain their food security. Therefore, the aim of this study is to explore the relationship between agricultural operational income and digital inclusive finance. This study evaluates the impact of digital inclusive financing on agricultural operational revenue in three different ways: the perspectives of coverage, usage, and the degree of digitalization. This study investigates the routes via which digital inclusive finance may have an influence on agricultural firms as well as the potential moderating impact of traditional financial development in order to better understand the mechanism underlying the effect.

Due to its rapid development of digital inclusive finance, China has drawn the attention of academics who are interested in learning how this development may impact food security. Previous research has examined how digital inclusive finance affects the expansion of farmland and the upgradation of agricultural equipment [26] as well as the improvement of agricultural green total factor productivity [22]. Concerning the effect of digital inclusive finance on Chinese enterprises, previous studies have found that digital inclusive finance can relieve the financing constraints of small and medium-sized enterprises [18,19,20], help boost entrepreneurship [27], and promote innovation [28].

There is an existing research gap concerning how digital inclusive finance impacts the operating income of agricultural enterprises in order to assure food security. By shouldering the roles of food processing, food market supply, and food waste reduction, agricultural enterprises have become a non-negligible component in the maintenance of food safety. Therefore, this study further explores the effect of digital inclusive finance on agricultural enterprises’ operating income with which to maintain food security. Approached the following angles, this work may be able to add to ongoing research. This study first establishes the beneficial impact of digital inclusive finance on food security from the standpoint of agricultural operations by examining the relationship between digital inclusive finance and agricultural operating revenue. Second, this study confirms the strength of the breadth of coverage and depth of usage while identifying the weakness in the degree of the digitalization of digital inclusive finance in terms of sustaining agricultural operating revenue. Third, this study discovers that digital inclusive finance has a beneficial impact on supporting financing for agricultural firms, reducing inventory, and encouraging innovation by examining the pathway whereby operating income is affected by this type of finance. Finally, this study discovers that traditional finance development is still required in order to fully utilize digital inclusive finance.

The rest of this paper is organized as follows. Section 2 reviews the existing literature relating to digital inclusive finance and food security. Then, based on the literature review, the hypotheses of this study are proposed. Section 3 lists the data and methodology employed in the study. Section 4 reports the empirical results of baseline regression, the mediating effect test, and the moderating effect test. Section 5 presents a discussion of the results and the study’s findings. Section 6 recapitulates the findings of this study and proposes political suggestions.

## 2. Literature Review and Hypotheses’ Development

### 2.1. Theoretical Review

The concept of food security was initially propounded by the Food and Agriculture Organization of the United Nations in 1974; since then, the concept has been revised, with the UN stating that every individual should have access to “sufficient, safe and nutritious” food at all times to satisfy their needs for an “active and healthy life”. Among the explanations of food security, the FAO emphasizes “economic access to nutritious food” [29]. Therefore, sustainable food production and a functioning food supply chain are essential for maintaining food security. As it is essential for human well-being, food production is also a major determinant of climate and environmental change [30,31]. Improving technology and reducing food waste can help reduce the environmental influence from the food system [32,33]. However, the financing constraints faced by most agricultural enterprises have restricted the realization of efficient food production and unimpeded sales in order to avoid food waste. Therefore, this study attempts to find a solution to the financing constraint of agricultural enterprises.

The theory of Modigliani and Miller (1958) holds that the external capital and internal capital of an enterprise can be completely replaced in a perfect capital market [34]. In this scenario, the investment behavior of an enterprise is only affected by their demand rather than financial conditions. However, with the problems of information asymmetry and agency, external financing costs can be higher than internal financing costs, meaning that financial conditions can always affect the investment behavior of enterprises. Based on the problem of information asymmetry, Greenwald et al. (1984), S. C. Myers and Majluf (1984), and S. Myers (1984) proposed the pecking order theory under an imperfect market [35,36,37]. The theory suggests that financing constraints are reflected by the difference in internal and external financing costs and are positively correlated with the degree of information asymmetry. Instead of the problem of information asymmetry, Bernanke and Gertler (1989) and Gertler (1992) investigated the agency problem, which also renders the cost of external financing higher than internal financing in an imperfect capital market [38,39].

Digital finance, which combines traditional financial services with digital technologies, can be used to quickly and affordably deliver correct information. In fact, it can efficiently address the issue of information asymmetry and lower transaction costs. The provision of more complex financial services, such as buying insurance, obtaining bank credit, and investing, is now much easier with the assistance provided by artificial intelligence, big data, cloud services, blockchain, and others. In terms of food supply, this aspect can be maintained smoothly, and food wastage can be reduced. This is reflected in the growth of sales in agricultural enterprises. As a result, the operating income of agricultural enterprises is likely to increase.

### 2.2. Empirical Review

Theoretically, it is expected that inclusive finance will relieve the problem of information asymmetry between transaction parties in financing activities. According to empirical studies, China has made considerable advancements in the area of food security but has also contributed significantly to environmental contamination through its excessive use of chemical pesticides and fertilizers [40]. Therefore, it is essential for China’s mode of agricultural development to move toward more effective methods of resource conservation and environmental sustainability [40,41]. Furthermore, the appropriate storage and smooth circulation of agricultural products are also vital for reducing food waste and enhancing food security [42]. To provide sufficient financial support, previous studies have reported that digital inclusive finance can support agricultural green development [12,22] and help relieve the financing constraints of small and medium-sized enterprises [15,17,18,19]. However, to the best of our knowledge, studies on the relationship between digital inclusive finance and agricultural operating income are non-existent. Therefore, this study aims to investigate the effect digital inclusive finance has on agricultural operating income by affecting financing supply, the sales of agricultural products, and agricultural research and development.

#### 2.2.1. Digital Inclusive Finance and Agricultural Operating Income

Agricultural enterprises can easily be excluded from traditional finance as they possess the characteristics of a long cycle, slow capital turnover, and high vulnerability to natural conditions [43]. Therefore, financing difficulties exist in China’s agricultural sector [44]. Since digital inclusive finance has the objective of increasing the provision of financial services to firms and individuals, it may relieve the financing constraints of agricultural enterprises in the three following manners. Firstly, it adopts a technical business mode to expand businesses in order to cover more regions and customers [18,19,20]. Secondly, it takes advantage of big data technology to solve the problem of information asymmetry and effectively reduce transaction costs [15,16,17]. Thirdly, by using information technology, agricultural enterprises can be offered more insurance products [45], thereby reducing credit risk and lowering interest rates [46]. Therefore, this study argues that digital inclusive finance can provide more financing support so that agricultural enterprises can realize greater operating income.

In addition to financing support, this study proposes that digital inclusive finance can support agricultural enterprises in terms of product sales through digital services. Unimpeded selling is also essential for ensuring sufficient food security. The disruption of an agricultural supply chain can result in the expiry of stored agricultural products, food wastage, and poverty among farmers [1,2]. However, in comparison to concerns relating to production, food selling has garnered less attention [47]. In reality, with supply chain disruption aggravated by the outbreak and spread of COVID-19, food cannot be delivered on time, thereby exacerbating the food crisis [48,49]. Therefore, an investigation into the development of measures to help liquidate agricultural inventories is relevant with respect to alleviating the global food crisis. Fortunately, the significant development of digital inclusive finance may ease this disruption in two ways. Firstly, with greater financing support, agricultural enterprises can adopt more advanced storage infrastructure and cold-chain transportation methods. Therefore, the perishability of agricultural products during storage and transportation can be further improved so that such products can resist market risks caused by supply chain disruption [50]. Secondly, digital technology provides convenient payment devices and affects consumption modes [13,14]. The expansion of online selling can effectively increase sales channels and accelerate inventory turnover. Therefore, this study argues that digital inclusive finance can help increase sales, which is indicated by inventory turnover, thereby increasing the operating income of agricultural enterprises.

Besides financing support and the promotion of sales, this study also attempts to explore whether digital inclusive finance can help improve agricultural product quality through supporting research and development. Since technological advancement is also recognized as a factor whereby agricultural productivity can be improved and income can be increased, by increasing investment in agricultural research and development (R&D), high-quality crops and an efficient mode of production can be realized [51,52]. This can improve agricultural operating income while also realizing cost-efficient and sustainable production and help sustain food security [53]. However, a low level of investment restricts the realization of potential research achievements [54]. Research activities have the characteristics of high uncertainty and asymmetrical information. Therefore, there is a greater likelihood of firms encountering financing difficulties in terms of investment into research and development [55,56]. There is also empirical evidence that financing constraints have impeded research and development investment by firms [57,58]. As digital inclusive finance may be effective for alleviating financing constraints, this study argues that digital inclusive finance can support the research and development (R&D) investment of agricultural enterprises and thereby increases operating income. Therefore, this study hypothesizes the following:

**Hypothesis 1** **(H1):**
*Digital inclusive finance positively affects agricultural operating income by increasing financing support, improving agricultural product sales, and supporting agricultural research and development.*


#### 2.2.2. Traditional Financial Development, Digital Inclusive Finance, and Agricultural Operating Income

As an innovative form that is based on traditional financial services, an industry’s digital inclusive finance performance may be influenced by traditional finance. Although it is presented in different forms, digital inclusive finance is rooted in traditional finance [59]. Some studies have suggested that digital inclusive finance complements traditional finance [60]. As innovation from traditional finance, the regional heterogeneity of digital inclusive finance may lie in the unbalanced development of regional traditional finance [61]. Meanwhile, the development of digital inclusive finance may also facilitate the promotion of regional traditional finance through a “technical spillover effect” [62]. Therefore, the moderating effect of regional traditional financial development cannot be ignored in this study.

**Hypothesis 2** **(H2):**
*Traditional financial development, together with digital inclusive finance, can help improve agricultural operating income.*


## 3. Data and Methodology

### 3.1. Data

The aim of this study is to investigate the effects of digital inclusive finance on agricultural operating income in order to food security. In order to achieve this target, this study focuses on small and medium-sized enterprises (SMEs), as digital inclusive finance is designed to relive the financing constraints of SMEs, which constitute more than 95% of Chinese enterprises. This study adopts the sample of agricultural enterprises listed in National Equities Exchange and Quotations (NEEQs) to represent agricultural SMEs in China as these enterprises are relatively small in size and are currently not qualified to go public with A shares. Enterprises listed in NEEQs are required to publicly provide audited financial statements, so their financial data are more reliable. Although they are relatively small in size, in comparison to those not listed in NEEQs, these enterprises are keen to finance and expand their businesses. They have supply chains covering local and surrounding areas and some even have a nationwide supply chain. Therefore, this sample should reflect the condition of the food supply chain in China. The sample period of 2011–2021 is long enough to observe the development of inclusive finance in China and reveal the challenges posed by the COVID-19 pandemic. Following the exclusion of enterprises with significantly poor business and omitted data, a total of 185 enterprises and 1068 company-year observations were collected. Their financial statements were obtained from the WIND database. The development of digital inclusive finance and other variables were determined from the prefecture-level data. Macro-economic variables were collected from city statistical yearbooks.

The development of digital inclusive finance is measured by the Peking University Digital Financial Inclusion Index of China (PKU_DFIIC). This index was constructed by the Digital Finance Research Center at Peking University and the Research Institute of Ant Group. Using a massive set of data relating to digital inclusive finance transactions, the PKU_DFIIC was constructed using the “analytic hierarchy process”. The index covers almost every region of China and is currently the most accurate index used by academics to measure the overall development and evolution of digital inclusive finance in China. Based on the concept of traditional financial inclusion, the research group considered specific features of digital financial services and constructed a digital financial inclusion index system according to three dimensions: breadth of coverage, depth of usage, and digitization level of inclusive finance. Digital financial inclusion was then determined by the aforementioned three dimensions and thirty-three specific indicators [13]. In 2022, the index was updated for the fourth time to cover provincial and prefecture-level indexes from 2011 to 2021.

### 3.2. Selected Variables

The core dependent variable of this study is agricultural operating income. Operating income per unit asset of sample enterprises is adopted to measure the average operating income per asset (AVGOI). From the perspective of agricultural enterprises, the sales of products can indicate the market supply of food. This study adopts operating income per asset rather than quantity of products sold because the selection of operating income as the dependent variable allows for the measurement of the quantity of products sold while also reflecting how much an enterprise can gain from its operations. The majority of agricultural products are life essentials with low price elasticity of demand. A high number of product sales does not necessarily lead to more income. Furthermore, operating income gained per asset is essential for the profit-making and debt coverage abilities of enterprises and can impact their sustainable operation, thus impacting food security in the long term. Therefore, this study focuses on the impact on operating income. In the robustness test, the net operating cash flow per asset is used as a substitute dependent variable to reflect the cash gain from selling by asset.

The core independent variable is the development level of digital inclusive finance, which is measured by the Peking University Digital Inclusive Finance Index (DIFI). In order to obtain a comprehensive understanding of the influencing mechanism, this study further investigated digital inclusive finance via three dimensions: breath of coverage (Breadth), depth of usage (Depth), and degree of digitization (Digital) using corresponding sub-items from the Peking University Digital Inclusive Finance Index.

This study adopts mediating variables to explore the channels through which digital inclusive finance affects agricultural operating income. Financing supply, inventory liquidity, and R&D investment are employed as mediating variables. Financing supply is the variable that is most directly linked with digital inclusive finance, and financing improvement can help agricultural enterprises expand their business such that they are closer to economic scale. Financing supply is measured by funds obtained from debt per unit total asset (Debt). Inventory liquidity is the variable that is potentially influenced by digital technology adopted in digital inclusive finance. Agricultural enterprises that are covered by digital inclusive finance may be able to sell more products on time and have a greater likelihood of increasing their operating income. Inventory liquidation is measured by inventory turnover ratio (INV). R&D investment can also be affected by digital inclusive finance. With the support of digital inclusive finance, agricultural enterprises can invest more in R&D activities, thereby improving their operational efficiency and product quality. As a result, more products can be efficiently produced, and high-quality products can be sold at higher prices. Production efficiency and increases in selling price can lead to increased operating income. Research and development (R&D) investment is measured by research and development investment per unit total asset (RDI). Employing these mediating variables can help us develop a better understanding of how digital inclusive finance affects agricultural operating income. Moreover, the mediating variables reflect the quantity and quality of products of agricultural enterprises and the efficiency of the supply chain with respect to selling perishable products in a timely manner. Therefore, these factors are also essential for maintaining food security.

As a type of financial innovation, traditional finance development can also impact the effect of digital inclusive finance on agricultural operating income. Digital inclusive finance can hardly be implemented if local financial institutions cannot enhance basic financial services. Therefore, in this study, traditional financial development is employed as moderating variable. The level of traditional financial development (FD) is measured by the loan balance of the current year/total local GDP of the current year of local financial institutions.

Table 1 shows the key variables of this study and Table 2 shows the descriptive statistics of the variables.

### 3.3. Methodology

In order to investigate the relationship between digital inclusive finance and the operating income of agricultural enterprises, pooled ordinary least square model is adopted. The specific model is as follows:
(1)$${\mathrm{AVGOI}}_{\mathrm{it}\;}=\;\alpha \;+{\;\beta \mathrm{DIFI}}_{\mathrm{it}}+\;\gamma \sum {\mathrm{Controls}}_{\mathrm{it}}+\varepsilon_{\mathrm{it}}$$
where AVGOI represents operating income gained per total asset, i represents enterprise, and t represents year. DIFI represents the digital financial inclusion index. ∑Controls represents the set of control variables and ε represents the residual term. In order to analyze the effect of digital inclusive finance in detail, the independent variable of DIFI has also been replaced by the following sub-dimensional indexes of digital inclusive finance: breadth of coverage (Breadth), depth of usage (Depth), and level of digitalization (Digital).

To further explore the mechanism whereby digital inclusive finance impacts the operating capability of agricultural enterprises, the following mediating effect model is adopted:
(2)$${\mathrm{AVGOI}}_{\mathrm{it}\;}=\;\alpha \;+{\;\beta \mathrm{DIFI}}_{\mathrm{it}}+\;\gamma \sum {\mathrm{Controls}}_{\mathrm{it}}+\varepsilon_{\mathrm{it}}$$
(3)$$M_{\mathrm{it}\;}=\;\alpha \;+{\;\beta \mathrm{DIFI}}_{\mathrm{it}}+\;\gamma \sum {\mathrm{Controls}}_{\mathrm{it}}+\varepsilon_{\mathrm{it}}$$
(4)$${\mathrm{AVGOI}}_{\mathrm{it}\;}=\;\alpha \;+{\;\beta \mathrm{DIFI}}_{\mathrm{it}}+{\;\delta M}_{\mathrm{it}}+\;\gamma \sum {\mathrm{Controls}}_{\mathrm{it}}+\varepsilon_{\mathrm{it}}$$
where M represents the set of mediating variables.

To establish whether traditional finance has a moderating effect on digital inclusive finance, the following moderating effect model is adopted:
(5)$${\mathrm{AVGOI}}_{\mathrm{it}\;}=\;\alpha \;+{\;\beta \mathrm{DIFI}}_{\mathrm{it}}+\;\gamma \sum {\mathrm{Controls}}_{\mathrm{it}}+\varepsilon_{\mathrm{it}}$$
(6)$${\mathrm{AVGOI}}_{\mathrm{it}\;}=\;\alpha \;+{\;\beta \mathrm{DIFI}}_{\mathrm{it}}+{\delta \mathrm{DIFI}}_{\mathrm{it}\;}{*\mathrm{FD}}_{\mathrm{it}}+{\delta \theta \mathrm{FD}}_{\mathrm{it}}+\gamma \sum {\mathrm{Controls}}_{\mathrm{it}}+\varepsilon_{\mathrm{it}}$$
where FD represents the level of traditional financial development.

## 4. Empirical Results

### 4.1. Analysis of Effects of Digital Inclusive Finance on Operating Income

Table 3 demonstrates that at the 1% significance level, digital inclusive finance increases agricultural operating income, with a coefficient of 1.631. This suggests that agricultural operating income increases by 1.631% for every 1% of growth in digital inclusive finance development. The F statistic suggests the entire model has good fitness.

As a means of further investigating the effect of digital inclusive finance on agricultural operating income, heterogeneity analysis was conducted to examine whether the effect differs between enterprises of different sizes and in different locations. The sample enterprises were divided into small and medium-sized enterprises based on the standards of the Ministry of Industry and Information Technology of China (MIIT). Columns 1 and 2 in Table 4 show the results. Digital inclusive finance was found to have a significant positive impact on the operating income of both small and medium-sized enterprises, with respective coefficients of 0.554 and 1.781. This study also examined the impact of digital inclusive finance on agricultural operating income in different functioning regions. The scheme for the division of the functioning regions originated from the 2001 grain distribution system reform in China. Based on natural resources and grain production conditions, 31 provinces (autonomous regions and municipalities) were divided into main production regions, main selling regions, and balanced regions. The results can be seen in columns three to five of Table 4. Interestingly, the impact digital inclusive finance has on agricultural operating income remained significant in the main selling regions and balanced regions but was insignificant in the main production regions. This may because agricultural production has been industrialized based on the advantageously developed natural resources of the main production regions. In addition, machinery and large-scale agricultural operation have already been adopted in the main production regions and many residents are reliant on agricultural output as their main form of income. As a result, the impact of digital inclusive finance on agricultural operation in the main production regions is not obvious. Meanwhile, the remaining significant impact of digital inclusive finance on agricultural operational income in the main selling regions and balanced regions indicates that the effect of digital inclusive finance is not limited to financing support but also includes agricultural products’ marketing and transportation.

### 4.2. Analysis of Digital Inclusive Finance Index on Sub-Dimensional Basis

To develop a more comprehensive understanding of the impact of digital inclusive finance, the sub-dimensional indexes of digital inclusive finance were analyzed with regression models. Table 5 demonstrates the results. Generally, the regression models have good fitness. With coefficients of 0.702 and 1.589, the regression coefficients for the breadth of coverage and depth of use in digital inclusive finance, respectively, are significantly positive. This indicates that increasing the coverage and usage of digital inclusive finance can increase agricultural operating income. Surprisingly, contrary to our expectation, the digitization level of digital inclusive finance was found to be insignificant in terms of its impact on agricultural operating income. This may be due to the following three reasons. Firstly, the effect of the digitization of digital inclusive finance on agricultural operation is restricted by insufficient financial infrastructure and technical hurdles in some regions [15,63]. Digitalization levels reflect the development and application of digital technology, and a high digitization level helps spread financial services in remote areas that have relatively few financial institutions. However, finance remains the crux of digital inclusive finance. Digital techniques can promote the spread of financial services, but digital inclusive finance will not be fully effective without actual improvements in coverage and usage. This result is consistent with a report from the Peking University Digital Financial Inclusion Index [13] regarding regional disparity, where digitalization has the smallest level of disparity due to its wide spread that is free from geospatial constraints. However, the level of regional disparity remains large in terms of the breadth of coverage and depth of usage. Secondly, lower financial literacy among rural residents may be a factor that limits the usage of digital functions required by digital inclusive finance [64,65]. Agricultural enterprises purchase raw agricultural products and reproduce them to deliver to consumers, but digital technology usage is limited as some farmers are unaware of how to connect with potential buyers via digital information platforms. This leads to agricultural enterprises needing to purchase raw agricultural products from intermediary agents at higher prices. The lack of digital technology usage restricts its effect on agricultural operating income. Thirdly, financial institutions may adopt discriminatory marketing tactics for digital inclusive finance products [66]. A higher level of digitization indicates higher credit risk and may be introduced to fewer agricultural enterprises. Limited usage of inclusive digital finance may further hinder the progress of digitization. Therefore, the improvement of the depth of usage and breadth of coverage are currently more effective for the stimulation of agricultural operating income.

### 4.3. Mediating Effect Analysis

This study argues that financing supply, inventory liquidity, and R&D investment have partial mediating effects. This is due, firstly, to the fact that digital inclusive finance can increase the amount of financing available to agricultural enterprises. So, the enterprises can increase the number of inputs in their operations. These inputs can be used to improve mechanization and expand businesses coach that they are closer to an economic scale [67,68]. Therefore, operating production can be increased more efficiently. Secondly, digital inclusive finance can support enterprises’ ability to update their storage and transportation equipment. Thus, agricultural products can be stored better before being sold, which results in less waste [50]. In addition, the use of a digital platform can incentive online consumption [14]. With a broader market for selling, inventory liquidity can also be improved. As a result, more products can be sold in time and thereby realize operating income. Thirdly, digital inclusive finance can support the investment of enterprises in research and development. As a result of progress in terms of R&D, the production of products with superior quality and in greater quantities is possible [51,52], and an increase in operating income can occur.

Table 6 reports the results of the mediating effect test. Generally, the regression models have good fitness. Column 1 to Column 4 demonstrate that digital inclusive finance can significantly increase operating income, financing supply, inventory liquidity, and R&D investment. Column 5 demonstrates that financing supply, inventory liquidity, and R&D investment have significant positive effects on operating profit. The empirical results are in line with expectations, confirming that financing supply, inventory liquidity, and R&D investment have partial mediating effects on improving operating income with the help of digital inclusive finance. The results indicate that digital inclusive finance can increase financing supply to agricultural enterprises. With more financing capital, these agricultural enterprises can more effectively expand their operational capacities, ultimately leading to an increase in operating income. In addition, digital inclusive finance can facilitate the promotion of agricultural product sales. This is reflected in an increase in inventory turnover, which improves operating income. Furthermore, with digital inclusive finance support, the R&D investment of agricultural enterprises can be increased, resulting in improved operating income.

### 4.4. Moderating Effect of Traditional Financial Development

This study argues that the development of traditional finance, as the basis of digital inclusive finance, can have a moderating effect. To explore the moderating effect of traditional financial development, first, a moderating effect test was conducted on digital inclusive finance. Table 7 demonstrates the results. Overall, the regression model has good fitness. The coefficient of DIFI × FD is significantly positive. This result indicates that traditional financial development has a positive moderating effect.

To enable a comprehensive exploration of the moderating effect, this study further conducted a moderating effect test on the sub-dimensional indexes of digital inclusive finance. The results are presented in Table 8. The coefficients of Breadth × FD, Depth × FD, and Digital × FD are all significantly positive. Surprisingly, the coefficient of Digital*FD is significantly positive, although digitization level is found to have an insignificant effect on agricultural operating income (as shown in Table 4). This suggests that although the digitization level itself does not lead to improved agricultural operating income, it can significantly promote agricultural operating income with the help of traditional finance development. This result demonstrates that to more effectively help digital inclusive finance improve agricultural operating income, the development of traditional finance is vital. The benefits of coverage breadth, usage depth, and digitization are all improving as traditional finance continues to develop.

### 4.5. Robustness Tests

In order to ensure the reliability of this study’s empirical results, we conducted a robustness test by substituting the dependent variable. Net operating cash flow is used to measure the operating income of agricultural enterprises. The regression of the digital inclusive finance index and sub-dimensional indexes are re-run with the substitute dependent variable. The results are shown in Table 9.

We also re-ran the mediating effect test and moderating effect test using the substituted dependent variable. In addition, the independent variable of DIFI that lagged for one time period is used in the robustness test of the mediating effect test. The robustness test’s results remain consistent. Due to spatial limitations, the results are presented in Appendix A
Table A1, Table A2 and Table A3

## 5. Discussion

Previous studies have reached the consensus that digital inclusive finance can relieve financing constraints by reducing information asymmetry. However, considering the natural risk in agricultural operations, it is still unclear whether digital inclusive finance can effectively support the agricultural sector [25,69]. By investigating the effect of digital inclusive finance (DIF) on the operating income of agricultural enterprises, this study provides evidence that, overall, digital inclusive finance can improve the operating income of agricultural enterprises and thus enhance food security. These findings fill the previously identified research gap by providing evidence that DIF can support agricultural operating income so as to enhance food security in China. Similar results were found in the studies by Lin et al. (2022) and Gao et al. (2022), wherein it was determined that digital inclusive finance can promote food security through the promotion of agricultural production in China [22,26]. Regarding disparities between agricultural functioning regions, the findings of this study are consistent with those obtained by Lin et al. (2022). The effect of digital inclusive finance on food security is prominent in the main selling regions and balanced regions but is insignificant in the main production regions. This may be due to the operations in agricultural main production regions already being mature and efficient. The results also indicate that digital inclusive finance may significantly promote the sale of agricultural products. Therefore, this study further investigated the impacts of DIF on inventory liquidation (indicating the quantity of sales) and R&D investment (indicating the quality of sales) as mediators in terms of influencing agricultural operating income.

As an innovation in financial services, this study provides evidence that digital inclusive finance can effectively increase financing supply to agricultural enterprises. This is consistent with the results obtained in the studies by Chen and Yoon (2022) and Lu et al., (2022) where it was found that digital inclusive finance can relieve the financing constraints of Chinese small and medium-sized enterprises [18,19]. However, this study delves further to investigate whether the effect of digital inclusive finance could further improve the operating income of agricultural enterprises. The results were significantly positive. As confirmed by previous studies, an increase in financing supply can help improve the capacity and efficiency of agricultural production [67,68]. Therefore, this study can confirm the intermediary role of financing constraints with respect to digital inclusive finance and agricultural operating income and concludes that digital inclusive finance can help increase agricultural operating income as a means of ensuring food security.

In addition to the mediating effect of financing supply, this study provides evidence that digital inclusive finance can help increase inventory liquidity. Through the use of information technology, digital inclusive finance can facilitate financing supply and payment and open up the online market. The findings further confirm that digital technology has a positive effect on the promotion of e-commerce, as was proposed by previous studies [13,14]. Based on previous studies, this study further concludes that digital inclusive finance can enhance food security. With the promotion of e-commerce, this study confirms that agricultural enterprises can sell perishable products more quickly. Furthermore, with greater financing supply, agricultural enterprises will have the necessary funds to upgrade their storage and transportation facilities [50], thus enabling them to better manage the risk of supply chain disruption. As a result, inventory liquidity can be increased, and food wastage can be decreased. Unimpeded sales are also vital for reducing food wastage and improving food security.

Whether increased financing can support R&D activities has been a controversial topic. This study provides evidence that sufficient financing supply supplemented by digital inclusive finance can increase the research and development (R&D) investments of agricultural enterprises. With more investment in R&D activities, both the quality and quantity of agricultural products can be improved [51,52]. Therefore, enterprises can gain greater operating income by upgrading their products and improving efficiency.

Finally, although the empirical results supported the notion that digital inclusive finance can significantly improve agricultural operating income, the sub-index of the digitalization level itself was found to have an insignificant effect on agricultural operating income, which indicates that digitalization alone is not adequate for the promotion of agricultural operating income. This result is also in accordance with the findings of Sun and Tang (2022). In their study of the effects of digital inclusive finance on sustainable economic growth in China, Sun and Tang (2022) also found digitization to be ineffective in terms of increasing sustainable economic growth in China [70]. We believe that this is because the support for financial services is currently insufficient for the digitization of DIF to effectively influence operating income. Therefore, this study further investigated whether traditional finance development could moderate the effect of digitization. The empirical test of this study concluded that with the help of traditional financial development, the influence digitization level of digital inclusive finance on agricultural operating income is significantly positive. Furthermore, the interactions between traditional financial development and the digital inclusive finance index and sub-dimensional indexes all have significantly positive effects. These results indicate that digital inclusive finance generated more significant effects with respect to supporting agricultural operations when combined with traditional financial development.

## 6. Conclusions

Based on the crucial role agribusiness plays in the food supply chain, this study explored the effect of digital inclusive finance on agricultural operating income to answer the following question: “can digital inclusive finance help enhance food security?”. The empirical studies shown herein employed a sample of agricultural enterprises in China listed in NEEQs from 2011–2021. The empirical results of this study find that digital inclusive finance can improve agricultural operating income. Concurrently, two of its sub-dimensions, namely, coverage breadth and usage depth, can significantly support agricultural operating income by itself. This study also proves that digital inclusive finance can support agricultural operating income by providing greater financing supply, increasing agricultural inventory liquidity, and supporting agricultural R&D investment. In addition, this study explores the significant role played by traditional financial development with respect to helping digital inclusive finance improve agricultural income. Particularly, with the help of traditional financial development, digitization level, which has no significant effect when operating alone, was found to be significantly effective in terms of increasing agricultural operating income.

The following suggestions are proposed based on the findings of this study. Firstly, the continuation of the development of digital inclusive finance in the agricultural sector will be effective for sustaining food market supply. Digital inclusive finance can effectively support operating input and help increase agricultural inventory turnover. Secondly, supporting research and development investment in the agricultural sector using digital inclusive finance is of great importance. In addition to food production efficiency, food qualification and diversification are important in order to enhance food security. Thirdly, at the current stage, the promotion of the breadth of coverage and breadth of use of digital inclusive finance would be more effective. Financial digitization is only effective when combined with traditional finance. This indicates that traditional finance still plays a crucial role, and its development should not be neglected when promoting digital inclusive finance.

There are limitations to this study in terms of data size. To certify the quality of the data, this study used financial data of agricultural enterprises listed in NEEQs and the Peking University Digital Financial Inclusion Index of China (PKU_DFIIC) to measure the development of digital inclusive finance. These are the most reliable data available, but the sample size of 185 enterprises is relatively small. Further studies examining the effects with a larger sample size or over a longer period of time may be conducted to yield further insights.

## Figures and Tables

**Table 1 ijerph-20-02956-t001:** Definition of variables.

Type	Variable	Definition
Dependent Variable	AVGOI	Average operating income; Operating income/Total assets.
	CFO	Substitute dependent variable for robustness test; Net operating cash flow/Total assets.
Independent Variable	DIFI	Logarithm of Digital Financial Inclusion index
	Breadth	Logarithm of Breadth of coverage
	Depth	Logarithm of Depth of usage
	Digital	Logarithm of Degree of digitization
Moderating Variable	FD	Traditional Financial Development; Local institutional loan balance/Local GDP
Mediating Variable	Debt	Funds obtained by Debt/Total assets
	INT	Inventory turnover ratio
	RDI	Research and Development Investment/Total assets
Control Variable	Size	Firm’s total assets
	Age	Years listed in NEEQs
	LNGDP	Logarithm of prefecture-level GDP in corresponding year
	Year	Dummy variable, with a value equal to 1 when belonging to the year; otherwise, the value is 0

**Table 2 ijerph-20-02956-t002:** Descriptive statistics of variables.

Variable	Observations	Mean	Std. Dev.	Min	Max
AVGOI	1068	0.812	0.977	0.000	14.230
CFO	1068	0.911	1.078	0.000	15.027
DIFI	1068	5.479	0.207	4.392	5.885
Breadth	1068	5.450	0.229	4.509	5.918
Depth	1068	5.446	0.248	4.338	5.870
Digital	1068	5.602	0.181	3.606	5.998
FD	1068	0.773	0.505	0.000	4.590
Debt	1068	0.174	0.186	0.000	1.520
INV	1068	0.016	0.049	0.000	1.000
RDI	1068	0.011	0.021	0.000	0.166
Size	1068	18.994	1.160	13.356	22.746
Age	1068	6.710	1.582	0.000	14.000
lnGDP	1068	11.042	0.624	2.809	12.070

**Table 3 ijerph-20-02956-t003:** Regression results of digital inclusive finance on operating income.

Variables	AVGOI
DIFI	1.631 ***
	(4.44)
Size	0.117 ***
	(4.92)
Age	−0.062 ***
	(−3.17)
lnGDP	0.038
	(0.63)
Year	Yes
Observations	1068
R-squared	0.074
F	5.970

t-statistics in parentheses. *** denote significance level at 1%.

**Table 4 ijerph-20-02956-t004:** The impact of digital inclusive finance on operating income according to different sizes of enterprises and functioning regions.

Variables	(1) Small Enterprises	(2) Medium Enterprises	(3) Main Production Regions	(4) Main Selling Regions	(5) Balanced Regions
DIFI	0.554 *	1.781 ***	0.709	8.50 ***	0.705 *
	(1.40)	(4.67)	(1.25)	(4.01)	(1.73)
Size	0.154 ***	0.086 ***	0.053 **	0.651 ***	0.061 ***
	(6.60)	(3.23)	(2.12)	(6.58)	(2.12)
Age	0.008	−0.059 ***	−0.072 ***	−0.117 **	−0.078 ***
	(0.32)	(−2.94)	−3.17)	(−2.05)	(−2.89)
lnGDP	−0.046	0.036	−0.123	0.050	0.122 *
	(−0.596)	(0.58)	(−1.16)	(0.33)	(1.94)
Year	Yes	Yes	Yes	Yes	Yes
Observations	1068	1068	1068	1068	1068
R-squared	0.607	0.067	0.040	0.259	0.100
F	5.720	5.150	2.05	4.16	2.51

t-statistics in parentheses. *, **, and *** denote significance level at 10%, 5%, and 1%, respectively.

**Table 5 ijerph-20-02956-t005:** Regression results of sub-dimensional digital inclusive finance index.

Variables	(1)	(2)	(3)
	AVGOI	AVGOI	AVGOI
Breadth	0.702 **		
	(2.46)		
Depth		1.589 ***	
		(6.44)	
Digital			0.332
			(0.84)
Size	0.112 ***	0.127 ***	0.109 ***
	(4.70)	(5.36)	(4.56)
Age	−0.062 ***	−0.058 ***	−0.058 ***
	(−3.13)	(−3.01)	(−2.97)
lnGDP	0.127 **	0.040	0.226
	(2.08)	(0.78)	(5.23)
Constant	−4.088 ***	−7.101 ***	−3.196 *
	(−3.49)	(−6.22)	(−2.06)
Breadth	0.702 **		
	(2.46)		
Year	Yes	Yes	Yes
Observations	1068	1068	1068
R-squared	0.062	0.092	0.057
F	4.940	7.620	4.53

Note: *, **, and *** denote significance level at 10%, 5%, and 1%, respectively.

**Table 6 ijerph-20-02956-t006:** Mediating effect.

Variables	(1)	(2)	(3)	(4)	(5)
	AVGOI	Debt	INV	RDI	AVGOI
DIFI	1.631 ***	0.162 *	0.041 **	0.019 **	1.065 ***
	(4.44)	(2.09)	(2.44)	(2.24)	(3.24)
Debt					0.510 ***
					(3.88)
INV					8.556 ***
					(14.09)
RDI					7.132 ***
					(5.87)
Size	0.117 ***	0.050 ***	0.000	−0.002 ***	0.107 ***
	(4.92)	(9.92)	(0.22)	(−4.36)	(4.75)
Age	−0.06 ***	−0.009 *	−0.002 *	−0.000	−0.041 **
	(−3.17)	(−2.12)	(−1.79)	(−0.83)	(−2.36
lnGDP	0.038	−0.011	0.012	0.002 *	0.011
	(0.63)	(−0.86)	(0.67)	(1.74)	(0.20)
Constant	−7.13 ***	−1.18 ***	−0.16 **	−0.062 *	−4.69 ***
	(−5.05)	(−3.99)	(−2.52)	(−1.92)	(−3.70)
Year	Yes	Yes	Yes	Yes	Yes
Observations	1068	1068	1068	1068	1068
R-squared	0.074	0.129	0.021	0.212	0.267
F	5.970	11.16	1.59	21.07	22.46

Note: *, **, and *** denote significance level at 10%, 5%, and 1%, respectively.

**Table 7 ijerph-20-02956-t007:** Moderating effect of traditional financial development.

Variables	(1)	(2)
	AVGOI	AVGOI
DIFI	1.631 ***	1.601 ***
	(4.44)	(4.42)
DIFI × FD		0.926 ***
		(3.31)
FD		0.213 ***
		(4.10)
Size	0.117 ***	0.129 ***
	(4.92)	(5.45)
Age	−0.062 ***	−0.063 ***
	(−3.17)	(−3.27)
lnGDP	0.038	0.027
	(0.63)	(0.44)
Constant	−7.127 ***	−6.865 ***
	(−5.05)	(−4.88)
Year	Yes	Yes
Observations	1068	1068
R-squared	0.074	0.101
F	5.970	7.400

t-statistics in parentheses. *** denote significance level at 1%.

**Table 8 ijerph-20-02956-t008:** Sub-dimensional moderating effect.

Variables	(3)	(4)	(5)	(6)	(7)	(8)
	AVGOI	AVGOI	AVGOI	AVGOI	AVGOI	AVGOI
Breadth	0.702 **	0.881 ***				
	(2.46)	(3.11)				
Breadth × FD		0.960 ***				
		(3.63)				
Depth			1.589 ***	1.433 ***		
			(6.44)	(5.81)		
Depth × FD				0.655 ***		
				(2.90)		
Digital					0.332	0.387
					(0.84)	(0.98)
Digital × FD						0.622 *
						(1.91)
FD		0.219 ***		0.176 ***		0.241 ***
		(4.20)		(3.38)		(4.63)
Size	0.112 ***	0.126 ***	0.127 ***	0.135 ***	0.109 ***	0.124 ***
	(4.70)	(5.30)	(5.36)	(5.72)	(4.56)	(5.17)
Age	−0.062 ***	−0.065 ***	−0.058 ***	−0.059 ***	−0.058 ***	−0.061 ***
	(−3.13)	(−3.35)	(−3.01)	(−3.06)	(−2.97)	(−3.10)
lnGDP	0.127 **	0.086	0.040	0.040	0.226 ***	0.217 ***
	(2.08)	(1.43)	(0.78)	(0.79)	(5.23)	(5.08)
Constant	−4.088 ***	−4.539 ***	−7.101 ***	−6.458 ***	−3.196 **	−3.249 **
	(−3.49)	(−3.91)	(−6.22)	(−5.62)	(−2.06)	(−2.03)
Year	Yes	Yes	Yes	Yes	Yes	Yes
Observations	1068	1068	1068	1068	1068	1068
R-squared	0.062	0.093	0.092	0.112	0.057	0.080
F	4.936	6.732	7.621	8.249	4.531	5.684

t-statistics in parentheses. *, **, and *** denote significance level at 10%, 5%, and 1%, respectively.

**Table 9 ijerph-20-02956-t009:** Robustness test of digital inclusive finance index and sub-dimensional index.

Variables	(1)	(2)	(3)	(4)
	CFO	CFO	CFO	CFO
DIFI	1.628 ***			
	(4.08)			
Breadth		0.630 **		
		(2.03)		
Depth			1.632 ***	
			(6.08)	
Digital				0.455
				(1.06)
RDI				
Debt				
INV				
Size	0.104 ***	0.099 ***	0.114 ***	0.096 ***
	(4.02)	(3.82)	(4.44)	(3.70)
Age	−0.071 ***	−0.071 ***	−0.067 ***	−0.067 ***
	(−3.35)	(−3.30)	(−3.20)	(−3.16)
lnGDP	0.071	0.170 **	0.066	0.255 ***
	(1.07)	(2.57)	(1.19)	(5.42)
Constant	−7.047 ***	−3.817 ***	−7.164 ***	−3.525 **
	(−4.59)	(−3.00)	(−5.77)	(−2.10)
Year	Yes	Yes	Yes	Yes
Observations	1068	1068	1068	1068
R-squared	0.067	0.056	0.085	0.053
F	5.412	4.471	6.949	4.243

t-statistics in parentheses. ** and *** denote significance level at 5%, and 1%, respectively.

## Data Availability

The datasets analyzed during the current study are not publicly available due to institutional restrictions but are available from the corresponding author on reasonable request.

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
