# Peer review of "Enhancing Food Security through Digital Inclusive Finance: Evidence from Agricultural Enterprises in China"

_ijerph, 2023, doi:10.3390/ijerph20042956_

Round 1
Reviewer 1 Report
The issue warrants investigation. Methods are appropriate and the research has been well conducted. However, the paper needs to be further elaborated. My main concerns are the following. First, the link between operating income of agricultural enterprises and food security needs to be better worked out. Second, the type of agricultural enterprise analyzed in the paper could be more precisely defined. See more details below on these two comments as well as some additional comments and suggestions.
· Overall, the paper needs to be further elaborated. Style and syntax need to be improved. Food-related vocabulary could be a bit more precise.
· The link between operating income of agricultural enterprises and food security is not at all clear to me. I would suggest better explaining and justifying the link. I just see the link between operating income and the profitability of firms in the current version of the paper. Furthermore, I would like to know how operating income growth indicates the quantity and quality of market food supply.
· There are a lot of types of agricultural enterprises. I would like to know more about the typology of the agricultural enterprises considered in this study. Defining this could help better focus the link mentioned in the previous comment. In addition, it is not clear to me whether the problem only concerns rural areas or both. Agricultural enterprises do not exclusively belong to rural areas, especially when the entire chain is considered. I would suggest further elaborating this point too.
· The paper’s aim needs to be better outlined. The paper’s objective encompasses more than what is set out in the introduction. It needs to be more comprehensive, so I would suggest further elaborating it in the introduction. The paper does not only examine the relationship between digital inclusive finance and agricultural operating income to better sustain food security in China. It also looks at the effect of breadth of coverage, depth of usage and degree of digitalization on agricultural operating income, the channels through which digital inclusive finance potentially impacts agricultural enterprises and the potential moderating effect of traditional financial development.
· Overall, the introduction needs to be improved. It does not clearly set out the problem behind the review, it does not point out the gaps in current knowledge and it does not demonstrate the need for research in the field. Addressing the three previous comments can help in doing so. I would also suggest further elaborating the paper’s contributions.
· I see little connection between the literature review and the introduction, even if the content is appropriate. The introduction should have somehow introduced the points elaborated in the literature review. I would suggest making an additional effort to better connect both sections. Some careful rewriting is also needed.
· Regarding the data, I would like to know whether the authors are considering the whole of China in the analysis. Regarding the prefecture-level digital inclusive finance measurement data, I would suggest first saying the authors are using prefecture-level data and then the source. Is it a balanced panel?
· Regarding the methods, after specifying the core model, I would suggest explaining the fact that models in which the dependent variables breadth of coverage, depth of usage and degree of digitalization are also being examined.
· Regarding the results of the mediating effect analysis, I would suggest explaining in more detail the effects of the mediating variables in the total outcome.
· The discussion could be somewhat more comprehensive.
Reviewer 2 Report
Thank you for the opportunity to read the article. Authors attempt to explore the impact of digital inclusive finance on food security by analyzing the effect of digital inclusive finance on operating income of agricultural enterprises in China. Although the paper attempts to address a timely issue, it is mired with problems that needs to be addressed. Following are my revision requests:
Introduction
Please clearly define research question or problem that is being addressed. Furthermore, the structure of paper is not outlined in the introduction section. The key terms such as “digital inclusive finance” and “agricultural operating income” should be defined for the reader.
In addition, please summarize the main contributions or findings of the study. It would be helpful to cite more references to previous research on the issue being addressed. It will help situate the study in the existing literature on food security and digital inclusive finance.
Literature Review and Hypothesis
The literature review and hypothesis part of the study might be enhanced by providing a clear summary of the existing research on the issue and by summarizing the key results and literature gaps. The relationship between the theoretical and empirical reviews may be clarified, and the literature study should conclude with a clear and succinct hypothesis.
The literature review might also benefit from a greater focus on China and references to new research on the subject. In addition, it would be beneficial to contextualize the study within the present state of the field and to provide further references to studies on food security and digitally inclusive finance in China. To this end, I am suggesting the following useful studies for the authors:
https://doi.org/10.3389/fenvs.2022.944156
https://doi.org/10.1016/j.technovation
https://doi.org/10.3390/agriculture11010049
https://doi.org/10.3389/fenvs.2022.918060
Methodology
Insufficient rationale is provided in the methods section for the selection of data sources and variables. Using the NEEQs sample may not be typical of all agricultural firms in China, and 185 enterprises is a rather small sample size. In addition, it is unclear how the Digital Inclusive Finance Index (DIFI) and its subdimensions were developed and if they are accurate metrics of digital inclusive finance. There is insufficient justification for the inclusion of finance supply, inventory liquidity, and R&D expenditure as mediating factors, and it is unclear how these variables connect to the primary study topic. The use of traditional financial development as a moderating variable is likewise not well supported, and its relationship to the other factors in the research is unclear. Clarity and rigor of the technique can be improved by providing more thorough descriptions and explanations of the data and variables. Finally, heading 3 and 3.3 is the same.
Results
According to Table 3, digital inclusive finance has a substantial beneficial effect on farm operational revenue. To gain a more thorough knowledge of the impact of digital inclusive finance on agricultural operations, it would be useful to explore the influence of digital inclusive finance on other financial variables, such as net profit and cash flow. In addition, it would be intriguing to determine if the impact of digital inclusive financing on operational income varies between small and large businesses, as well as across businesses in various locations.
In Table 4, the results demonstrate that both the breadth and depth of digital inclusive financing have a considerable beneficial influence on agricultural operational income, whereas digitalization level has no effect. It remains to investigate the underlying causes of this result, such as potential hurdles to the adoption of digital technology in rural regions and disparities in financial literacy. Can authors do further analysis on this dimension?
Table 5's mediating analysis gives some insight into the processes via which digital inclusive finance influences agricultural operational income, but authors should see whether these effects persist in a more robust study with a bigger sample size or over a longer period of time.
Discussion
This section is too short. A comparison has not been made with previous studies which does not effectively situate the article within existing scholarly debate. Authors need to enhance the discussion keeping in view these points.
Conclusion
Please mention the limitations of the study and expand on the future research directions.
Round 2
Reviewer 1 Report
The paper has been improved and most of the comments have been addressed satisfactorily. However, I would still suggest improving style and syntax and agriculture/food-related vocabulary.
Author Response
Thanks for your suggestion. This paper has been proof-read by a native speaker and revisions relating to English expression and style have been done by authors accordingly. Please refer to the changes in manuscript for details.
Reviewer 2 Report
I have no further comments.
Author Response
Thanks for your respond.